Parental thermal conditions affect the brain activity response to alarm cue in larval zebrafish

Sourisse Jade M. 1 2
Semmelhack Julie L. 3
Schunter Celia celiaschunter@gmail.com 1
1 The Swire Institute of Marine Science, School of Biological Sciences, The University of Hong Kong , Hong Kong SAR , China
2 Marine and Environmental Sciences Centre, Laboratório Marítimo da Guia, Faculdade de Ciências, Universidade de Lisboa , Cascais , Portugal
3 The Division of Life Science, The Hong Kong University of Science and Technology , Clearwater Bay, Kowloon , Hong Kong
Nowak Barbara
Electronic publication date: 2024 Oct 10
Publication date: 2024
Volume: 12
Electronic Location ID: e18241
Received 2024 Aug 9; Accepted 2024 Sep 15
Copyright: ©2024 Sourisse et al.
Copyright year: 2024
Copyright holder: Sourisse et al.
License: This is an open access article distributed under the terms of the Creative Commons Attribution License, which permits unrestricted use, distribution, reproduction and adaptation in any medium and for any purpose provided that it is properly attributed. For attribution, the original author(s), title, publication source (PeerJ) and either DOI or URL of the article must be cited.
License URL: https://creativecommons.org/licenses/by/4.0/

Keywords: Olfactory bulb, Climate change, Forebrain, Neurotransmission, Neurons

Funding: University of Hong Kong General Research Fund 17300721 NSFC Excellent Young Scientist Award AR225205 Jade M. Sourisse and this study were funded by the start-up of CS from the University of Hong Kong. The study was funded by a General Research Fund (17300721) and the NSFC Excellent Young Scientist Award (AR225205) to Celia Schunter. There was no additional external funding received for this study. The funders had no role in study design, data collection and analysis, decision to publish, or preparation of the manuscript.

==============================
Temperature is a crucial factor affecting the physiology of ectothermic animals, but exposure to elevated temperature during specific life stages and across generations may confer fish resilience through phenotypic plasticity. In this study, we investigate the effects of developmental and parental temperature on brain activity response to an olfactory cue in the larval zebrafish, Danio rerio. We exposed parents during reproduction and their offspring during development to control (28 °C) or elevated temperature (30 °C) and observed the response of the larval telencephalon to an alarm cue using live calcium imaging. Parental exposure to elevated temperature decreased the time till maximum brain activity response regardless of the offspring’s developmental temperature, revealing that parental thermal conditions can affect the excitability of the offspring’s neural circuitry. Furthermore, brain activity duration was affected by the interaction between parental and offspring thermal conditions, where longer brain activity duration was seen when either parents or offspring were exposed to elevated temperature. Conversely, we found shorter brain activity duration when the offspring were exposed to the same temperature as their parents, in both control and elevated temperature. This could represent an anticipatory parental effect influencing the offspring’s brain response to match the parental environment, or an early developmental effect occurring within a susceptible short time window post-fertilization. Overall, our results suggest that warming can alter processes involved in brain transmission and show that parental conditions could aid in the preparation of their offspring to respond to olfactory stimuli in a warming environment.

Introduction

Temperature is arguably the principal environmental factor that can influence fish performance (Brett, 1971; Johnston & Dunn, 1987) and has known effects on a variety of behaviours such as locomotion, food search and boldness (Fukuhara, 1990; Stoner, Ottmar & Hurst, 2006; Biro, Beckmann & Stamps, 2010) and brain function (Beltrán et al., 2021). Olfaction is one sensory modality in fish that is crucial for behaviours such as feeding, migration and predation escape (v. Frisch, 1938; Hara, 1975). It is mediated by olfactory sensory neurons (OSNs) in the olfactory epithelia, which converge in the telencephalon in a region called the olfactory bulb (Laberge & Hara, 2001). From the olfactory bulb, second order projections to other telencephalic olfactory areas provide input to neuronal pathways involved in behavioural responses such as escape or freezing (Jesuthasan & Mathuru, 2008; Miyasaka et al., 2009). The olfactory circuitry is sensitive to higher temperatures and shows warming-mediated hyperexcitability in fish (Flerova & Gdovskii, 1975; Døving & Belghaug, 1977) indicating potential implications for the persistence of olfactory triggered behavioural responses under global warming. However, it is unclear how far thermal conditions can impact olfactory processing in the fish brain (Tigert & Porteus, 2023).

One way by which fish may show resilience to thermal effects is phenotypic plasticity, which is the capacity of an individual to adjust its phenotype to changing conditions without altering its genetic constitution (Crozier & Hutchings, 2014; Fox et al., 2019). Throughout development, specific windows of susceptibility exist during which organisms are more responsive to external conditions (Burggren & Mueller, 2015), such as early embryonic stages in fishes (Mueller et al., 2015; Flynn & Todgham, 2018; Melendez & Mueller, 2021). For instance, temperature during fish embryogenesis is known to influence later metabolic performance and behaviour into adulthood (Jonsson & Jonsson, 2014; Jonsson & Jonsson, 2018). Furthermore, a plastic response to global warming may occur not only within, but across generations when parents alter the phenotype of their offspring via non-genetic inheritance, named intergenerational plasticity (O’Dea et al., 2016; Fox et al., 2019). In fish, parental exposure to elevated temperature can provide increased acclimation to their offspring (Donelson et al., 2012; Shama et al., 2014). Parental effects of warming have been shown to affect the brain, including expression of genes expression involved in neuromuscular junction development (Bernal et al., 2022) and hormonal pathways (Veilleux, Donelson & Munday, 2018). Despite the importance of these acclimation mechanisms to temperature elevation, it is not well understood how such within and intergenerational effects of warming may affect the offspring’s brain function, due to the complexity of the vertebrate brains.

To investigate the potential parental contributions to thermal acclimation in the fish brain as it senses its environment, we can make use of the zebrafish model. The neuronal circuitry governing olfactory-mediated processes has been extensively described in this model species (Friedrich, Jacobson & Zhu, 2010) thanks to the engineering of genetically encoded calcium indicators that allow brain activity to be observed at the larval stage, when the brain has functional yet less complex neuronal networks than the adult stage (Kettunen, 2012). Furthermore, temperature conditions experienced by parents have been shown to confer metabolic compensation in their offspring as a transgenerational response in this species, as routine metabolic rates of the offspring were reduced at stressful temperatures with biparental early life exposure to fluctuating thermal conditions (Massey & Dalziel, 2023). Finally, as the olfactory fear response is innate in zebrafish (Jesuthasan & Mathuru, 2008), it can be directly observed without the need for learning experiences to occur. Antipredator behaviour in zebrafish can be olfactory-mediated: if they perceive an olfactory alarm cue coming from an injured conspecific in their surrounding environment, they reduce or even suppress their locomotion activity (Speedie & Gerlai, 2008). Therefore, we can observe an innate olfactory response of ecological relevance in the larval stage.

In our study, we investigate the effects of developmental elevated temperature on the brain activity response of zebrafish larvae to an alarm olfactory cue, the Conspecifics Alarm Cue (CAC). Since warming was shown to shorten antipredator escape latencies (Domenici et al., 2019), we hypothesize that elevated thermal conditions will impact the neuronal circuitry by provoking hyperexcitability, as it was previously reported in other fish species. Furthermore, we also assess the relative influence of parental thermal conditions and early embryonic temperature exposure to elevated temperature on the offspring’s brain activity response to CAC. We expect that previous exposure to elevated temperature will alter brain activity patterns in response to CAC of offspring reared under near-future predicted conditions. By characterizing thermally induced brain activity changes in response to CAC, we aim to determine how elevated temperature will affect the environmental perception of fish through the olfactory neuronal circuitry.

Methods

Portions of this text were previously published as part of a preprint (https://www.biorxiv.org/content/10.1101/2024.06.02.597016v1).

Animals, housing, and temperature exposure

Zebrafish breeders (five females, one male; Tg(elavl3:Has.H2B-GcaMP6s) strain) were obtained from the Hong Kong University of Science and Technology zebrafish husbandry facilities. This zebrafish strain is unpigmented and possesses a genetically encoded calcium indicator, allowing to show the calcium ion levels increase in the brain during neuronal activity. The parent breeders were housed in recirculating systems (80 × 37 × 32 cm), with a 14/10 h light-dark cycle. They were fed twice a day ad libitum with TetraMin flakes. pH and nitrate levels were measured weekly using a Seven2GO portable pH meter (Mettler Toledo) and a HI97728 nitrate photometer (Hanna Instruments), respectively. Parent breeders were first reared in control conditions (28 °C) for 46 days, during which fertilized eggs were collected to constitute an offspring group from control parental conditions (P28). Then, parents were exposed to elevated temperature (30 °C, reached over a day) and were bred after seven days to collect fertilized eggs belonging to an elevated temperature group (P30), for which warming was experienced during spawning and early embryogenesis until the end of the cleavage period of development (Kimmel et al., 1995). We selected the control temperature as it is the optimal rearing temperature of zebrafish in laboratory settings (Aleström et al., 2020). We chose the treatment temperature as +2 °C as this variation of temperature can already alter molecular (Lahiri et al., 2005) and behavioural responses of larval zebrafish (Abozaid, Tsang & Gerlai, 2020; Sourisse et al., 2023), furthermore their neural activity can be impaired before reaching the critical thermal limit (Andreassen et al., 2022). Temperature was measured, adjusted, and recorded every 60 s with heaters (Schego) and a STC-1000 Thermostat (Elitech). Temperatures experienced by the parents differed significantly before and during the thermal exposure (Wilcoxon test, p-value = 1. 743 ×10−8): the treatment temperature over the experimentation period was 29.8 ± 0.2 °C, whereas the mean control temperature was 27.7 ± 0.2 °C (Table S1).

For both groups of offspring according to parental spawning temperature (P28 or P30), fertilized eggs at the end of the cleavage period of development (Kimmel et al., 1995) were collected in the morning and then placed in Petri dishes filled with Danieau’s solution embryo medium (17 mM NaCl, 2 mM KCl, 0.12 mM MgSO4, 1.8 mM, Ca(NO3)2, 1.5 mM HEPES). They were housed in in DSI-060D incubators (Digisystem). Offspring from both parental temperature groups were themselves randomly divided into two thermal groups, either control (28 °C, O28) or elevated (30 °C, O30) temperature at the time of embryo collection. Temperature inside the incubators was automatically adjusted every 60 s. The embryo medium was changed daily. Embryos were reared under a 14/10 h light-dark cycle and from five days post fertilization (dpf) onwards they were fed a larval diet (Zeigler Bros) ad libitum daily until seven dpf. Temperatures between offspring treatments differed significantly (Wilcoxon test, p-value = 3. 883 ×10−10): the “treatment” temperature over the experimentation period was 30.1 ± 0.1 °C, whereas the mean control temperature was 28.1 ± 0.2 °C (Table S1). This study was carried out in approval of the Committee on the Use of Live Animals in Teaching and Research (CULATR) of the University of Hong Kong (#5614-21 and #22-257).

Brain imaging data acquisition

At 7 dpf, larvae were embedded in low-melting agarose inside a small Petri dish (Ø= 45 mm, h = 16 mm) with the anterior part of their head –therefore the olfactory epithelia –freed from agarose. The Petri dish was filled with six mL of medium and let to habituate for two to three hours. After habituation to embedding, larvae were placed under a confocal laser scanning microscope at a room temperature of 26 °C (LSM 980; Carl Zeiss). The confocal microscope recorded brain activity from the top using an excitation light of 488 m (FITC, Oregon Green) in the telencephalic region where OSNs converge into the olfactory bulb and other areas that mediate olfactory responses. Larvae were first exposed to 0.5mL of Danieau solution as a control introduced in the Petri dish by using a 2.5 mL syringe. After a 5 min interval, they were then exposed to 0.5 mL of Conspecifics Alarm Cue (CAC), a substance that is known to trigger an alarm response such as reduced or suppressed locomotion in larvae (Jesuthasan & Mathuru, 2008; Lucon-Xiccato et al., 2020). For each subject larva, donor zebrafish larvae were chosen randomly among larvae reared together with the subject and were sacrificed by head concussion to produce the CAC, at a concentration of four donors/mL and homogenizing their bodies in water with a sterile mortar and pestle, between 4 to 5 min before cue exposure time (Sourisse et al., 2023). This CAC concentration value, although higher than another used in previous studies (Lucon-Xiccato et al., 2020) was chosen to ensure the odorant stimulation to be strong enough to consistently elicit a brain activity response in all individuals. Cue exposure started after the confocal microscope captured a first frame of baseline brain activity. A total of 37 larvae were then used to record the brain activity response to CAC exposure (for the P28-O28 group, n = 8; for the P28-O30 group, n = 10; for the P30-O28 group, n = 9; for the P30-O30 group, n = 10), with 8 to 10 individuals per group chosen to account for individual variability. The confocal microscope recorded a total of 20 pictures per run, with each pictures shot every 1.65 s. Therefore, the acquisition of the brain activity response for each individual lasted 33 s.

Statistical analyses

Stack time series acquired were opened in Fiji v 2.14.0 (Schindelin et al., 2012) and stacks were aligned using the SIFT plug in (Saalfeld & Tomancák, 2008). To measure fluorescence, the telencephalon was manually defined as a discrete Region of Interest (ROI) in each frame. Its plane area was directly measured from the aligned stack acquisition (μm2) as a proxy of brain development (Table S2). Differences in fluorescence (ΔF) were then calculated as follows: the baseline fluorescence value (F) of the ROI was chosen as the lowest value during the first 1-3 frames of the time series, which corresponds to a period of low activity. To calculate ΔF throughout the acquisition, F was subtracted to the corresponding ROI’s fluorescence value at any given time following the baseline frame (ΔF). To ensure that increased brain activity recorded in larvae was due to CAC and not just any stimulus into the chamber, values of ΔF and ΔF/F over imaging acquisition time were used to compare runs performed on the same larva (control vs CAC): exposure to CAC produced a curve of positive values of ΔF over time whereas in the case of the control, ΔF would not produce a positive peak but decrease in ΔF over time due to fluid addition (Fig. 1 as example). Then, ΔF/F values obtained for CAC in the four different thermal groups were used to assess the effect of parental or offspring temperature on individuals brain activity response to CAC. Specifically, several metrics were statistically compared in R (R Core Team, 2018) such as the maximum ΔF/F value in the run (max ΔF/F), the time at which max ΔF/F occurred (max ΔF/F time) and the brain activity response length (defined as duration for which ΔF>0; Table S2). For max ΔF/F, one outlier was removed (Grubbs test, p-value = 0.019) before comparison. A two-way ANOVA was chosen when the response variable showed homogeneity of variances as per tested with a Bartlett’s test, otherwise a Scheirer-Ray-Hare test was performed.

Figure 1 Telencephalon mean fluorescence intensity (A) and ΔF/F (B) over time during control (blue) or Conspecifics Alarm Cue (CAC, red) exposure and brain activity during control (C) or CAC (D) exposure.

(A–B) Dots correspond to the measured values and lines plot smoothed conditional means; (C–D) the colour bars plot fluorescence intensity for each pixel; oe = olfactory epithelium; L = left; R = right; Te = Telencephalon; (C) corresponds to frame 1 whereas (D) corresponds to frame 4 of the acquisition.

Results

Results are reported as mean ± standard deviation (STD) values. We hypothesized that warming would alter the brain activity response of zebrafish larvae responding to CAC. Surprisingly, there was no effect of parental nor offspring temperature on the maximum brain activity level max ΔF (Fig. S1), nor on max ΔF/F (Fig. 2) which accounts for individual variances in fluorescence baseline levels in the brain. This means that neuronal response in the forebrain did not increase with parental nor offspring exposure to elevated temperature during the olfactory response.

Figure 2 Mean telencephalon calcium fluorescence ratio (ΔF/F) over time (frame) in zebrafish larvae exposed to Conspecifics Alarm Cue (CAC).

Parents that produced these offspring experienced either control (P28) or elevated temperature (P30) conditions; larvae were reared in either control (O28) or elevated temperature (O30) conditions; dots represent mean values whereas ribbons around solid lines correspond to standard deviation.

However, there was a significant effect of the parental spawning temperature on max ΔF/F time (Scheirer-Ray-Hare test, p-value = 0.0097; Fig. 3A). Larvae of parents who bred at control temperature reached their maximum brain activity at 9.53 ± 3.13 s (n = 18) whereas larvae born from parents experiencing elevated temperature reached maximum activity earlier, at 7.29 ± 1.68 s (n = 19).

Figure 3 Mean time of maximum calcium fluorescence (time of max ΔF/F; (A) and mean duration (B) of the telencephalon response to CAC in zebrafish larvae exposed to Conspecifics Alarm Cue (CAC).

Parents that produced these offspring experienced either control (P28) or elevated temperature (P30) conditions; larvae were reared in either control (O28) or elevated temperature (O30) conditions; stars (**) in (A) indicates the significant difference between offspring from the P28 group and that of the P30 group.

While there was no effect of developmental temperature on the time of max ΔF/F itself (Scheirer Ray Hare test, p-value = 0.852) nor any interaction effect between parental spawning and offspring thermal conditions (Scheirer Ray Hare test, p-value = 0.934), development in elevated temperature had an effect on the variability of max ΔF/F time (Bartlett test, p = value = 0.0123). Offspring reared at elevated temperature (n = 20) reached their maximum brain activity level at different times compared to offspring reared at control temperature (n = 17) for which maximum brain activity levels were mostly reached around 8.44 ± 2.9 s.

Although the brain response duration was not affected by parental spawning temperature (two-way ANOVA, p-value = 0.278), nor by offspring thermal exposure (two-way ANOVA, p-value = 0.686), there were interactions between these factors (two-way ANOVA, p-value = 0.022), suggesting that developmental warming has a different effect on zebrafish larvae brain activity depending on the thermal conditions experienced either during early embryogenesis and/or by their parents at time of reproduction (Fig. 3B). Larve reared at control temperature from parents exposed to the same temperature had a brain activity response duration of 20.83 ± 3.63 s (n = 8). This response duration tended to increase to 24.75 ± 5.5 s in larvae from the same reproduction conditions but developed at elevated temperature (n = 10), however it was not significant (Student’s t-test, p-value = 0.089). Longer lasting brain activity response was also not significant (Student’s t-test, p-value = 0.234) in offspring of parents reproduced at elevated temperature but reared in control conditions (duration = 23.65 ± 5.6 s, n = 10). Nevertheless, the brain response to CAC of offspring from parents reproducing and reared at elevated temperature lasted 18.48 ± 7.2 s (n = 9), which is more similar to the response in control fish.

We verified that the telencephalon area (11 010.65 ± 1111.89 µm2; Table S2) did not vary due to parental spawning temperature (two-way ANOVA, p-value = 0.782), nor offspring temperature (two-way ANOVA, p-value = 0.648) nor the interaction between those two factors (two-way ANOVA, p-value = 0.257; Fig. S2). This suggests that the differences observed are not due to physiological development accelerated by elevated temperature in the forebrain.

Discussion

We explored how warming experienced within and between generations affects the neuronal response of zebrafish larvae as they sense an alarming olfactory stimulus. We found that parental spawning and early embryogenesis in elevated temperature accelerated the time of maximal activity response. Furthermore, brain activity duration was affected by the interaction between parental spawning and offspring developmental thermal conditions.

Alarm cues provoke neural activity patterns in the telencephalon, as well as reduced mobility in larvae (Jesuthasan et al., 2021) and bottom dwelling and freezing in adults (Diaz-Verdugo et al., 2019). A change in the telencephalic response to CAC caused by exposure to warmer temperature could therefore lead to modifications of specific behaviours related to predator avoidance or escape (Speedie & Gerlai, 2008). Contrary to our expectations, development in elevated temperature alone did not significantly affect the time to reach maximum brain activity and only marginally affected the response duration to CAC. This is likely due to larvae developing in elevated temperature over seven days, contrary to an acute exposure in perch and char where hyperexcitability was reported with elevated temperature (Flerova & Gdovskii, 1975; Døving & Belghaug, 1977). However, larvae from parents reproducing at elevated temperature reached their maximum brain activity response earlier than offspring arising from reproduction at control temperature. Although there is previous evidence that offspring can inherit neural activity traits from their parents as they sense olfactory stimuli (Liu et al., 2017), our results show that neuron excitability is also increased by the thermal environment of their parents. Furthermore, it occurs through different mechanisms than the known direct effect of water warming on axon conductance (Flerova & Gdovskii, 1975; Døving & Belghaug, 1977), as we observed a faster brain activity maximum regardless of the offspring’s developmental temperature.

Additionally, there are complex effects of temperature on brain signal duration depending on the time of exposure. The interaction between parental and offspring exposure to warming would produce a different effect according to whether parental and offspring temperatures would match: brain activity would tend to last longer if parental and offspring temperatures differed, but when both parents and offspring were exposed to elevated temperature, a shortening of brain activity duration was observed, which was similar to control. This may represent an anticipatory parental effect, which consists in adjusting the offspring’s phenotype to the parental thermal environment, under the assumption that the parental environment corresponds to that of the offspring (Burgess & Marshall, 2014; Bale, 2015). Since “restoration” of brain activity duration was only observed when both parents and offspring were kept at the same elevated temperature, exposure to parental conditions may rescue normal brain functions when they would otherwise be affected as seen in another ectotherm species (Arai et al., 2009), but provided that the parental thermal environment is a reliable predictor of the offspring’s rearing temperature (Donelson et al., 2018). For stenothermic aquatic species, temperature experienced during parental reproduction may correspond well to the temperature experienced by the next generation (Somero, 2010; Logan & Buckley, 2015), however zebrafish in the wild are exposed to a wide range of seasonal thermal variation (Spence et al., 2008; López-Olmeda & Sánchez-Vázquez, 2011). Overall, elevated temperature experienced during the time of reproduction affects the brain activity response of their offspring during olfactory stimulation, which could have important potential implications on olfactory-mediated behaviour. However, as one of the parental effects is a compensation of a brain activity response prolongation, the impact of thermal conditions experienced by the offspring may be limited owing to exposure of the parents.

There are different pathways through which brain activity response could be changed after parental spawning in elevated temperature. First, such modifications could be non-genetic inheritance from either the paternal, maternal or both origins involving molecular regulation of neurodevelopment (Colson et al., 2019; Chan et al., 2020). Epigenetic changes that are transmitted to the offspring through the germline notably constitute a mechanistic basis of parental anticipatory effects (Öst et al., 2014; Rechavi et al., 2014). However, another potential mechanism may be that elevated temperature at reproduction directly acts on gametes or even non-sperm components of ejaculate, as seen in another fish species where sperm thermal environment influenced post-hatching performance (Läinen et al., 2018). In the latter case, cellular stress responses to warming in various components of the semen (Lane et al., 2014; Menezo et al., 2016) could also be responsible for epigenetic modifications (Immler, 2018) causing the observed changes in the offspring. Finally, a faster time of maximum brain activity could be a phenotypic plastic response expressed by the offspring itself triggered at their very early embryonic stage, due to their exposure to parental thermal conditions then. In other fishes, an increase in incubation temperature during gastrulation provoked changes in hatchling phenotype, revealing a critical window of development (Mueller et al., 2015; Melendez & Mueller, 2021) which shows that thermal experience during early embryogenesis provides relevant information about later conditions in fish’s life (Fawcett & Frankenhuis, 2015). Here, parental environmental conditions could influence the offspring directly after fertilization, during the short time before fertilized eggs were exposed to their developmental temperature during their week of incubation. Therefore, whether through parental effects, or abiotic effects on the gametes, sperm or fertilized eggs, modifications leading to changes in neuron activity are caused by the parental thermal environment, although our study does not allow us to pinpoint which mechanism(s) prevail.

A possible target of temperature-driven modifications in the zebrafish offspring may be processes involved in excitatory synaptic transmission. In particular, the dopaminergic system may be involved in intergenerational effects of warming on zebrafish brain activity: dopaminergic neuron populations are found in the larval zebrafish telencephalon (McLean & Fetcho, 2004). Furthermore, dopaminergic neurotransmission influences odour responses (Bundschuh et al., 2012) and dopamine modulates neuron firing (Schärer et al., 2012; McGregor & Nelson, 2020). Finally, the dopaminergic system can be affected in a transgenerational way through modified expression of dopaminergic receptors (Yu et al., 2021), which is also sensitive to warming in fish developing stages (Giroux, Gan & Schlenk, 2019). Here, epigenetic modification of dopaminergic receptors expression in the telencephalon caused by parental temperature conditions could increase neuron excitability in the offspring,. Another possibility is that parental exposure to elevated temperature could change molecular processes involved in inhibitory transmission, such as GABAergic neurotransmission. GABA is expressed as an inhibitory neurotransmitter in the majority of interneurons in the olfactory bulb (Miyasaka et al., 2013). In fish, proteins and genes involved in the transport and functioning of GABA were found altered within and across generations after exposure to another environmental stressor, ocean acidification, with consequences on signalling and antipredator behaviour (Schunter et al., 2016; Williams et al., 2019). Furthermore, elevated temperature has the potential to alter GABAergic-mediated behaviours (Clements, Bishop & Hunt, 2017). Here, warming experienced by the parents could affect GABAergic interneurons in a way that would reduce their inhibitory input on the olfactory circuitry. Overall, our results support a model in which neural excitability in the telencephalon of zebrafish larvae is affected by elevated temperature, with several possible and non-exclusive mechanisms of non-genetic inheritance at play either within or across generations.

The adaptive value of such parental conditioning effects conferred to the offspring is not clear, and it is often difficult to assess whether the response to parental environments is adaptive (Donelson et al., 2018; Candolin & Rahman, 2023), where changing one’s phenotype can buffer against effects of environmental change (Jonsson, Jonsson & Hansen, 2022). Although elevated temperature experienced as embryos was shown to have persistent positive effects on performance at elevated temperature for zebrafish adults (Scott & Johnston, 2012), such developmental temperature acclimation cannot always be expected as beneficial (Leroi, Bennett & Lenski, 1994; Huey et al., 1999). When it comes to plasticity observed at the brain level, warming in goldfish also caused hyperexcitability and these neural changes resulted in a faster escape response, but decreased the directionality of the escape (Szabo et al., 2008), suggesting that warming could have mixed effects on antipredator behaviour. Therefore, it is unclear whether parental conditioning effects on the time of maximum brain activity observed in our study would prove advantageous to the offspring when temperatures increase. Nevertheless, previous research on other vertebrates showed that timing of olfactory processing is important for odorant recognition, as perception depended on activation latencies of brain regions and cells, especially the cells that are activated earliest (Chong et al., 2020). This suggests that faster activation of the telencephalon caused by the parental environmental conditions could influence the perception of CAC and potentially the antipredator behaviour triggered by it (Speedie & Gerlai, 2008). Furthermore, considering that predators of prey fish are for the most part ectotherms themselves, faster telencephalic activity could help the offspring respond accordingly to quicker attacks from their predators, also influenced by elevated temperature as seen in other fish species (Allan et al., 2015). Finally, one may view as adaptive the other parental effect observed in our study, which is a restoration of brain activity duration when both generations experience elevated temperature. Since it maintains the phenotype observed in control conditions. Unlike purely developmental exposure to elevated temperature, it may prevent deleterious effects of prolonged brain activity response to olfactory stimulation, such as excitotoxicity, a degeneration of neurons resulting from of excessive excitation (Mattson, 2017).

In summary, our findings show an effect of elevated thermal conditions on developing zebrafish, reaching maximal telencephalic activity sooner than their control counterparts when exposed to an olfactory alarm cue. Furthermore, the thermally induced prolongation of neural activity could be mitigated if parents reproduced at such elevated temperature, suggesting potential parental anticipatory effects or early developmental plasticity in fish. These effects, particularly during sensitive early life-history stages, may serve to enhance persistence and adaptability, provided that parental conditions accurately predict the offspring’s environment.

Supplemental Information

Supplemental Information 1 Maximum fluorescence in zebrafish larval telencephalon exposed to Conspecifics Alarm Cue (CAC) according to their parents’ and their thermal exposure

Parents that produced these offspring experienced either control (P28) or elevated temperature (P30) conditions; larvae were reared in either control (O28) or elevated temperature (O30) conditions

Supplemental Information 2 Telencephalon plane size (μm2) of zebrafish larvae according to their parents’ and their thermal exposure

Parents that produced these offspring experienced either control (P28) or elevated temperature (P30) conditions; larvae were reared in either control (O28) or elevated temperature (O30) conditions

Supplemental Information 3 Experimental conditions experienced by parents, their offspring and water chemistry parameters throughout the experiment (1) andtelencephalon activity measurements performed on zebrafish larvae (2)

Supplementary Table 2: each row corresponds to an acquisition; “Run ID” refers to the order of fish measurements processing within one experimental date; “Cue” refers to the olfactory cue administered during the acquisition (control solution W, or alarm cue CAC); “F1” to “F20” are ΔF measurements in frames 1 to 20

Supplemental Information 4 ARRIVE Guidelines checklist

We thank Yan Chit Kam who contributed to the daily care of breeding animals and Helen Leung who helped with confocal imaging and all the members of the lab for support. We thank the reviewers who have helped us improve this manuscript with their suggestions.

Additional Information and Declarations

Competing Interests

Author Contributions

Animal Ethics

Data Availability

The authors declare there are no competing interests.

Jade M. Sourisse conceived and designed the experiments, performed the experiments, analyzed the data, prepared figures and/or tables, authored or reviewed drafts of the article, and approved the final draft.

Julie L. Semmelhack conceived and designed the experiments, authored or reviewed drafts of the article, and approved the final draft.

Celia Schunter conceived and designed the experiments, authored or reviewed drafts of the article, and approved the final draft.

The following information was supplied relating to ethical approvals (i.e., approving body and any reference numbers):

Committee on the Use of Live Animals in Teaching and Research (CULATR) of the University of Hong Kong.

The following information was supplied regarding data availability:

The brain fluorescence measurements are available in the Supplementary Tables.

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
