# Peer review of "Parental thermal conditions affect the brain activity response to alarm cue in larval zebrafish"

_PeerJ, doi:10.7717/peerj.18241_

## Round 0.1 · original submission · Minor Revisions

Please address all the comments from the reviewers. In particular, please remove any comments overinterpreting your results. If results are not statistically significant there is no difference (whatever appears to look different on the graph is only by chance) and null hypothesis of no effect cannot be rejected so it is essential that the results which have p-value higher than 0.05 are not reported or discussed as "different" or "trends" as they are only trends by chance. For example, one of these reported "trends" has a p-value of 0.089 and another a p-value is 0.234. These should not be reported or discussed, unless further analyses are done and their results provided, for example effect size (please see Cohen Statistical Power Analysis for the Behavioral Science, published by Taylor and Francis Group or Visentin et al 2019 https://onlinelibrary.wiley.com/doi/full/10.1111/jan.14283).

Furthermore, please ensure that Materials and Methods are clear and provide all information requested by the reviewers.

·

Basic reporting

Thermal conditions experienced by parents and during early development of the offspring can have major ecological, physiological and behavioural effects on the offspring later in life (review in Jonsson et al. 2022 Q Rev Biol). These effects may have a significant effect on the offspring fitness, although evidence of the adaptive advantage is still meagre.

In this MS, the authors show that thermal conditions of both parents and during early development of offspring affects the response time to an alarm substance of the offspring. The mechanism making this possible is unknown as is the possible adaptive value of this. However, for me it seems reasonable that offspring anticipating living in a warmer environment should respond sooner to a poikilothermic predator, as their attacks should be faster.

The MS is well written with a well-defined research question filling a relevant and timely gap in our knowledge. I would have liked that the authors had explained the ecological and evolutionary relevance of it more. I see the relevance of the research question in relation to earlier findings, but I think that the authors should also think more about how the hypothesized response would be beneficial to the animal under natural conditions. To be relevant, it should affect the animal under natural conditions, and more thinking about this would make the research more interesting and relevant.

The metods are well described and the authors know and use the physiological literature well. They could have included some more on effects of thermal effects on metabolic and swimming rates, to better make readers aware that it is important that prey species respond sooner in a warmer environment.

Underlying data are provided, and the use of statistics is good. The figures are fine. I feel that this MS will make a nice contribution to Peer J after a minor revision.

Experimental design

The experimental design is good. There are few parental fish, but else, I find it well described. This is a rigorous investigation of high technical standard described in sufficient detail.

Validity of the findings

No comment

Reviewer 2 ·

Basic reporting

Some editing on word choice throughout would improve the flow of the paper. Additionally, certain sentences are awkward to read and could be improved.
Examples:
Line 33, the word "crucial" is used twice in one sentence, six words apart.
Line 55; "Parental exposure", but then not making it clear what the parents are exposed to. I know you mean temperature, but should be stated.
Line 72: "Simplest form" sounds awkward and is a bit unclear. Do you mean the complexity of the behavioral response is simple?

Experimental design

Some concerns about the experimental design, albeit minor, I would like some clarification on.
Overall, I am unsure if only a 2 degree C increase in temperature is enough to truly invoke the changes the authors claim it can. I read the cited works, but zebrafish have a CTMax closer to 40 C, and 2 degree variation is something they would commonly experience now. Due to this, I am unsure if 30 C (+2C) can illicit the degree of plasticity you claim. And since temperature fluctuations of 2 degrees over seven days is something zebrafish experience now, I am not sure if it is potentially the best model design for future warming as you make it out to be.

Additionally, if a +2C change from rearing conditions can have lasting changes, does the same apply for a decrease in temperature by the same amount? All fish were tested at 26 C, which is -2C from control and -4C from treatment conditions. Would this decrease in temperature act as a confounding variable?

Line 123: I believe the values were swapped, it says the treatment was 28.1C, and the control is 30.1C.

Also some concern with the use of conspecific alarm cue and preparation. CAC are notoriously hard to use because of their short longevity, and begin to degrade very rapidly, as well as being hard to quantify and standardize. On line 141, the authors state that the CAC was prepared in "5 minutes or less". How large of a variability was there in this? Was there a set range of time, for example between 4 and 5 minutes? Or was the collection much quicker for some, around a minute, and others longer, closer to 5 minutes? Clarifying this would help. Additionally, the CAC was collected from 4 donor fish each time. Was the combined mass of the donor fish (approximately) the same size for each trial? At 7 days post fertilization, there can be some variation in mass, and using larger fish for one trial could mean a larger concentration of CAC was used as an odorant.

Validity of the findings

No comment.

Additional comments

No comment.

Reviewer 3 ·

Basic reporting

In line 80-81, the authors need to explicitly justify why they made this hypothesis, as it is currently not made clear to the reader. Additionally, while the authors refer to the fact that their hypothesis was not supported in lines 179-183 of the results, there is no mention of this in the discussion. The authors should at least briefly comment on this result in the discussion, with reference to the underlying theory that led to this hypothesis, discussing potential explanations for why this may not have been the case.

The following minor comments may also aid in improving the overall clarity of the manuscript:

Line 19-22: This sentence is a bit confusing to understand – consider rephrasing to something more along the lines of “Furthermore, brain activity duration was affected by the interaction between parental and offspring thermal conditions, where longer brain activity duration was seen when either parents or offspring were exposed to elevated temperatures. Conversely, shorter brain activity duration was seen when offspring were exposed to the same temperature as parents, regardless of whether it was the control or elevated.”

Line 31-33: Some specific examples of known effects of temperature on behaviour and brain function would help illustrate this point. Also, there should be a citation provided for brain function, given the claim that effects on brain function have previously been established.

Line 33: Avoid using the word “crucial” twice in one sentence.

Line 57: The word “were” should be replaced with “have been”.

Line 59: I think the word “these” instead of “those” would make the sentence read better.

Line 60-62: The phrasing here is difficult to follow – perhaps instead: “it is not well understood how such within and intergenerational effects of warming may affect the offspring’s brain function, due to the complexity of the vertebrate brains.”

Line 69: Again, “were” should be replaced with “have been”.

Line 69-71: It would be helpful to elaborate more about the specifics of what this study showed.

Line 72: There should be a comma between “zebrafish” and “it can be observed…”.

Line 86: This sentence would read slightly better if “at determining” is replaced with “to determine”.

Line 95: It would be worth specifying the amount of TetraMin flakes that were provided to the fish, or stating that the fish were fed ad libitum. This also applies to line 121 when referring to larval diet.

Line 99: To me it seems unnecessary to say “later named”, instead of just provided the assigned code in brackets. This also applies each time a treatment group name is defined.

Line 104: A citation should be provided regarding the optimal rearing temperature of zebrafish in laboratory settings.

Line 193: “Reaching” should be “reached”.

Line 198: Replacing “interaction effect of those two factors” would read better if reworded as “interactions between these factors”.

Line 201-202: The phrasing here is a little clunky – perhaps rephrase as “Larve reared at control temperature from parents exposed to the same temperature had…”.

Lines 217-257: This paragraph is far too long. The content discussed should be broken into smaller paragraphs to aid flow and improve readability. The same is true for a number of other paragraphs in the discussion.

Line 253: The word “right” is unnecessary in “directly right after fertilization”.

Line 257: This should be reworded as “does not allow us to pinpoint”.

324-326: This sentence is quite difficult to follow and needs rephrasing.

Experimental design

no comment

Validity of the findings

The authors have put quite a lot of emphasis on some outcomes that, as stated in their results, are not statistically significant. Greater caution is needed in interpreting these results given the reported p-values. One of these reported trends has a borderline significant p-value of 0.089 – for this one I think it is worth commenting on, although the authors should still be careful not to overstate this finding too much in the discussion. However, a p-value is 0.234 is too high to warrant much emphasis, and any interpretations that rely on this trend being biologically relevant need to be reassessed and toned down. This is particularly true for lines 286-296 in the discussion, which draws strong inferences from these trends that lack statistical significance.

Additional comments

The study is well designed, high quality and has interesting results. My main concern is that the authors may have overemphasised some results that did not have statistical significance, which could be misleading to readers. Additionally, I have provided more minor suggestions that I believe will improve the clarity and overall readability of the manuscript. With these adjustments, I think the manuscript will be a valuable contribution to PeerJ.

---

## Round 0.2 · Minor Revisions

Thank you for making the revisions. I am concerned that p=0.089 is considered "marginally significant" even it is 1.78 times greater than p normally accepted as significant. Please include what you considered as statistically significant in Materials and Methods. If it was p<0.05 then p=0.089 is not marginally significant. It is not significant. If you accepted greater p value as significant please clarify in Materials and Methods and explain the reasons for raising significant p value from the normally accepted and how it could have affected your results. I would really appreciate if you could please respond to the Editor's comments from this and previous revision.

---

## Round 0.3 · accepted · Accept

Thank you for addressing all the comments. As they were my comments I made the editorial decision without involving the reviewers and I am happy with the current version. The manuscript is now ready for publication.